# Complete Hydatidiform Mole with Lung Metastasis and Coexisting Live Fetus: Unexpected Twin Pregnancy Mimicking Placenta Accreta

**DOI:** 10.3390/diagnostics13132249

**Published:** 2023-07-03

**Authors:** Hera Jung

**Affiliations:** Department of Pathology, CHA Ilsan Medical Center, CHA University School of Medicine, Goyang 10414, Republic of Korea; elledriver2008@gmail.com

**Keywords:** complete hydatidiform mole, gestational trophoblastic disease, twin pregnancy

## Abstract

Twin pregnancy with a complete hydatidiform mole and coexisting fetus (CHMCF) is an exceedingly rare condition with an incidence of about 1 in 20,000–100,000 pregnancies. It can be detected by prenatal ultrasonography and an elevated maternal serum beta-human chorionic gonadotropin (BhCG) level. Herein, the author reports a case of CHMCF which was incidentally diagnosed through pathologic examination without preoperative knowledge. The 41-year-old woman, transferred due to preterm labor, delivered a female baby by cesarean section at 28 + 5 weeks of gestation. Clinically, the surgeon suspected placenta accreta on the surgical field, and the placental specimen was sent to the pathology department. On gross examination, focal vesicular and cystic lesions were identified separately from the normal-looking placental tissue. The pathologic diagnosis was CHMCF and considering the fact that placenta accreta was originally suspected, invasive hydatidiform mole was not ruled out. After radiologic work-up, metastatic lung lesions were detected, and methotrexate was administered in six cycles at intervals of every two weeks. The author presents the clinicopathological features of this unexpected CHMCF case accompanied by pulmonary metastasis, compares to literature review findings, and emphasizes the meticulous pathologic examination.

## 1. Introduction

Complete hydatidiform mole (CHM) is one of the gestational trophoblastic diseases originating from fertilization of an empty ovum by a sperm and can be invasive or metastatic [1]. In the early stage of CHM, clinical presentations including vaginal bleeding and a snowstorm appearance of the ultrasound lead to the detection of the disease [2]. Additionally, maternal serum beta-human chorionic gonadotropin (BhCG) level elevation also assists the prenatal diagnosis [2]. Of itself, CHM does not have the fetal part; however, twin pregnancy with a complete hydatidiform mole and a coexisting fetus (CHMCF) has been documented in about 1 in 20,000–100,000 pregnancies and the precise diagnosis of CHMCF can be delayed [3]. Herein, the author reports a case of unexpected CHMCF referred to the pathology department with a clinical impression of placenta accreta in a preterm labor.

## 2. Case Presentation

A 41-year-old G3-P1 multigravida woman, with 28 weeks and 4 days of gestation, was admitted to the author’s institution because of preterm labor and a need for treatment in the neonatal intensive care unit (NICU). The patient had an obstetric history of dilatation and evacuation due to spontaneous abortion 4 years previously at GA (gestational age) 11 weeks, and 3 years previously she delivered a female baby weighing 3.2 kg at GA 41 weeks. The transfer record from an outside hospital presented a low-lying placenta with a suspicion of abruption and pelvic examination result of 3 cm and 50% effacement. In the first ultrasonographic finding of the present institute, the fetus was small for the gestational age (27 + 3 weeks, 5.6 percentile) and due to the fetal position, the heart and extremities were not checked. Along with the low-lying placenta, hypervascularity and high blood flow in the subplacental area to the uterine fundus were identified. Other findings also included bridging vessels and multiple irregular lacunae within the placenta in the color Doppler ultrasound. The previous history of evacuation, maternal age, and ultrasonographic findings suggested the possibility of placenta accreta or placenta increta (Figure 1A). Moreover, there was a 4.4 cm × 3.2 cm × 2.5 cm sized mixed echoic lesion in the cervical canal, and a blood clot was suspected (Figure 1B).

After removing the blood clot with a speculum, the membrane bulged, and the length of the cervix became 0 cm with U-shaped funneling. Although magnesium sulfate (Magnesin) and ritodrine (Lavopa) were administrated, labor pain persisted every 5 to 8 min with 30–80 torr. As ultrasound findings suggested placenta accreta, the obstetrician obtained informed consent for cesarean section with the possibility of uterine artery embolization and hysterectomy in case of excessive bleeding. An emergent cesarean section was conducted on the day after admission (at GA 28 + 5 weeks). On the surgical field, the uterus was slightly dextrorotated and enlarged to term size. Bilateral ovaries and fallopian tubes were grossly normal in size and shape. The clear amniotic fluid was noted. A living female baby weighing 1030 gm with an Apgar score of 7 (at 1 min) and 8 (at 5 min) was delivered in the left occiput transverse position. Intraoperatively, the uterus showed no obvious distension over the placental bed and the surface was clear without gross neovascularity. After an initial trial of manual removal of the mildly adherent placenta, bleeding was present but controlled after an intravenous Pitocin (10 unit) injection. Therefore, no further procedure was initiated. Although the operative findings were not fully sufficient for a placenta accreta spectrum (PAS) diagnosis, the preoperative ultrasound and the experienced clinician’s suspicion did not exclude placenta accreta, so the specimen was sent to the pathology department. The patient tolerated the entire procedure well and recovered in a stable condition. On gross examination at the pathology department, the placental specimen consisted of a discoid-shaped placental tissue, weighing 728 gm and measuring 23 cm × 16 cm × 2 cm. The umbilical cord inserted centrally 5 cm apart from the nearest margin and was measured 35 cm in length and 2.2 cm in diameter. On section, it had two arteries and one vein. The amniotic membrane was semitransparent. The fetal surface of the chorionic plate was smooth and semitransparent. The maternal surface was covered by intact cotyledons with blood clots and there were also separated multiple fragments of vesicular tissue, measuring up to 13 cm × 11 cm in aggregates (Figure 2A). Considering the heterogeneous gross findings and clinical suspicion of placenta accreta, sections were obtained at variable portions of the specimen. Microscopic examination demonstrated two distinct areas of villi: (1) hydropic large villi with peripheral trophoblastic hyperplasia and cistern formation; and (2) relatively small normal villi (Figure 2B). The areas of hydropic villi had massive necrotic changes, more than about 80%, and in the viable area, the enlarged villi had an internal cistern formation and circumferential trophoblast hyperplasia with often cytologic atypia (Figure 2C). Villous stromal cells and cytotrophoblasts of hydropic villi area were negative for p57 immunohistochemical staining, the marker for the maternally expressed gene CDKN1C (p57KIP2) (Figure 2D). The histologic and immunohistochemical results were consistent with complete hydatidiform mole. Meanwhile, p57 showed retained expression in normal-looking villi (Figure 2E) and there was multiple defined proliferation of capillary vessels with surface trophoblastic proliferation, consistent with chorangiomas (Figure 2F). The size of the largest chorangioma was measured to 0.5 cm. These whole pathologic findings indicated an unexpected twin pregnancy with CHMCF.

Because the placenta was removed manually during surgery, there was no clear distinctive border. Additionally, the surgeon originally suspected placenta accreta and only placenta was sent for pathologic examination without any uterine tissue, so the possibility of invasive hydatidiform mole was not excluded in the clinical context. The final pathologic report was twin pregnancy with CHMCF and indicated the possibility of invasive hydatidiform mole, so a BhCG level check and radiological work-up for excluding residual or metastasizing lesions were recommended. The BhCG level at 15 days after delivery was 1325 mIU/mL. There was no previous BhCG data because an emergent section was performed. Chest computed tomography (CT) revealed variably sized nodules in both lungs, indicating hematogenous metastasis (Figure 3). Brain CT was normal and abdominopelvic CT showed postpartum uterine enlargement, fatty liver, and borderline hepatosplenomegaly. 

Six cycles of methotrexate injection were administered every two weeks. After each cycle, the BhCG level gradually decreased (399–33.9–7.2–2.4–1.2–0.4 mIU/mL). The last BhCG level was 0.2 mIU/mL at five months after delivery and follow-up CT confirmed no evidence of recurrence or metastasis in the chest and abdominopelvic cavity. The preterm baby had respiratory distress syndrome but improved and was discharged with a 2260 gm weight after two months of NICU care.

## 3. Discussion

Gestational trophoblastic disease is categorized by putative trophoblastic cells of placental origin; chorionic villous trophoblasts and intermediate trophoblasts [1]. Of them, hydatidiform mole originates from chorionic villous trophoblasts and is divided into complete, partial, and invasive types [1]. Among them, the pathogenesis of CHM is associated with the presence of a paternal-only genome [1]. The majority (about 80–90%) of cases are caused by duplication of the paternal haploid genome, detected as genome-wide homozygosity (46, XX), and the rest are produced by dispermy, resulting in heterozygosity (46, XX or 46, XY) [1,4]. Rarely, inherited mutation of NLRP7 or KHDC3L have also been identified as causes of familial biparental CHM [1]. Overexpression of the paternal genome leads to failure of normally balanced placental and fetal development [4]. As a result, on microscopic examination, CHM is characterized by enlarged chorionic villi with a cistern formation. Circumferential trophoblastic hyperplasia with cytologic atypia is also a usual finding and p57 immunohistochemical staining is negative in villous stromal cells and cytotrophoblasts. In CHM, fetal parts are normally absent. However, CHMCF cases have been steadily reported with low prevalence (1/20,000–100,000) [3,5,6,7,8,9,10,11,12]. The median gestational age at diagnosis of CHMCF is 15–16 weeks, and the delivery or termination is performed in 21–24 weeks [3,9]. Clinical symptoms include vaginal hemorrhage, preeclampsia, and hyperthyroidism [3]. According to the largest review article of CHMCF by M. Suksai et al., more than half of patients (118/206, 57.28%) have hemorrhage and initial BhCG levels range from 1048 to 2,460,000 mIU/mL with a median level of 367,747 mIU/mL [9]. Ultrasonography can also help the diagnosis, demonstrating snowstorm appearance and a heterogeneous, echogenic mass with cystic appearance [13]. Despite the traditional recommendation for termination of the pregnancy, several studies suggests that the risk of gestational trophoblastic neoplasia after CHMCF is not significantly increased with continuation of the pregnancy [9,10]. M. Suksai et al. reported that 37.86 % (78 of 206) were delivered successfully compared to 22.33% (46 of 203) of miscarriage or intrauterine fetal death, stillbirth, and neonatal death [9]. A better prognosis is statistically associated with the lower prevalence of antenatal maternal complications, such as pregnancy-induced hypertension (PIH), hyperthyroidism (HTD), and hyperemesis gravidarum (HG) [9]. An initial serum BhCG level less than 400,000 mIU/mL is also known as a favorable predictive factor for live births [9].

In the present case, the patient had not been diagnosed with CHMCF before and there was no serum BhCG results due to emergent admission. However, the absence of PIH, HTD, and HG might have contributed to the successful delivery. Placenta accreta was the initial clinical impression when the placental specimen was referred to the pathology department. Distinct vesicular tissues were observed on gross examination by the pathologist, so the hidden molar pregnancy obscured by a normal living fetus could be properly diagnosed. In this pregnancy, the patient was confirmed to be pregnant while living abroad but had entered South Korea during the second trimester due to the COVID-19 pandemic (patient’s delivery date: 9 May 2022). The limitations on hospital visits during the pandemic period of COVID-19 are considered as a possible explanation for the delay in the diagnosis of CHMCF. The significant amount of necrosis might also be another factor that made prenatal diagnosis difficult [14]. Meanwhile, the incidence of chorangioma is 1% and associated with an increased risk of pregnancy complications, including polyhydramnios and preterm delivery [15]. Known risk factors of chorangioma include a maternal age over 30 years, maternal hypertension, twin pregnancy, maternal smoking history, and living at high altitude [15]. In the present case, the placenta of the normal living fetus had multiple chorangiomas, and two factors (maternal age and twin pregnancy) might have contributed to the development of the tumors. As multiple chorangiomas can have some overlapping ultrasonographic findings with molar pregnancy, cautious radiologic reading is also required [15]. Clinically, degenerating molar tissue can mimic placenta accreta [7]. In this case, the clinician’s suspicion of placenta accreta helped the pathologic diagnosis of CHMCF with a possible invasive or metastasizing lesion. As a result, metastatic lesions that might have been missed were found, and the patient had effective chemotherapy. A lack of previous hospital information including ultrasonography and initial serum BhCG was a limitation in this case.

Additionally, the author attempted to compare this case with previous reports of CHMCF with lung metastasis. The Medline database was thoroughly searched using the PubMed retrieval service. The keywords used were “complete hydatidiform mole and surviving coexistent twin”, “complete hydatidiform mole twin metastasis”, “complete mole twin lung”, “complete mole twin pulmonary”, “complete mole fetus lung”, and “complete mole fetus pulmonary”. Among the studies, the cases without English publication were excluded. A total of 20 cases were collected, as those with an unspecified metastasis site and limited clinical information were omitted. Including the presented case, the clinical information from the 21 cases is displayed in Table 1. The median maternal age was 34 years. Some pregnancies were achieved by IVF (in vitro fertilization) (3 cases), hMG/hCG (human menopausal gonadotropin/human chorionic gonadotropin) therapy (1 case), and ICSI (intracytoplasmic sperm injection) (1 case). Most of the collected cases were diagnosed by prenatal BhCG or radiologic examination. Only one case in 1982 was diagnosed on delivery [16]. Thirteen cases attempted delivery including cesarean section, but in two cases, the infants died within a few hours. The detection of pulmonary metastasis was usually made after termination/delivery. Only four cases were detected before delivery at a mean GA of 25 weeks (17–32 weeks). Compared to the previous studies, the present case demonstrates the importance of pathologic examination. In most of the cases, coexisting complete hydatidiform mole was recognized in the first or second trimester, unlike this case. It is exceptional that the hidden complete hydatidiform mole and multiple lung metastases that could be harmful to the patient were diagnosed through the accurate pathological examination.

## 4. Conclusions

In summary, an unexpected twin pregnancy with CHMCF and extrauterine metastasis which were clinically mimicking placenta accreta is reported. Such uncommon cases can be detected by pathological examinations, so it should always be conducted out of caution even for usual specimens. Furthermore, if placenta accreta is suspected, it is worth considering a serum BhCG check when only limited clinical information is available, such as in this case. 

## Figures and Tables

**Figure 1 diagnostics-13-02249-f001:**
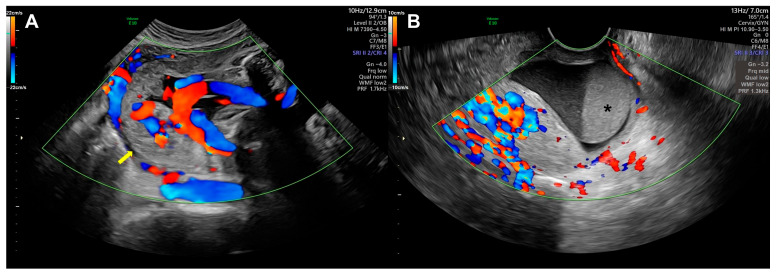
Ultrasonographic findings after admission. (**A**) Placenta with hypervascularity and high blood flow in subplacental area (yellow arrow); (**B**) Blood clot in cervical canal (asterisk).

**Figure 2 diagnostics-13-02249-f002:**
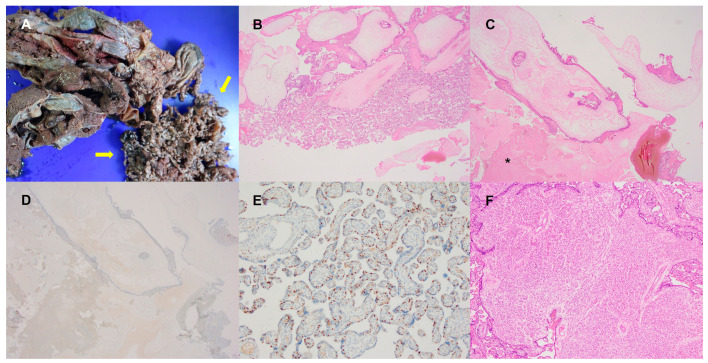
Gross and microscopic findings of CHMCF. (**A**) Separately identified small vesicles on gross examination (yellow arrow); (**B**) Two groups of villi: hydropic villi with cistern formation and relatively small normal-looking villi (12.5×); (**C**) Complete hydatidiform mole area with massive necrosis (asterisk) (12.5×); (**D**) Negative p57 immunohistochemical staining of complete hydatidiform mole (12.5×); (**E**) Positive p57 immunohistochemical staining of normal area (100×); (**F**) Chorangioma (40×).

**Figure 3 diagnostics-13-02249-f003:**
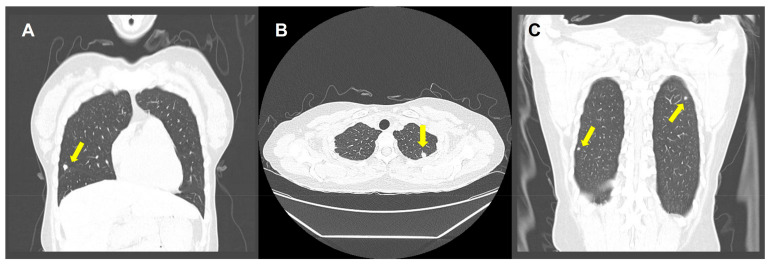
Chest computed tomography (CT) highlighting multiple pulmonary metastasis (yellow arrows). (**A**) Metastatic lesion in right middle lobe on coronal view; (**B**) Metastatic lesion in left upper lobe on axial view; (**C**) Metastatic lesions in both lobes on coronal view.

**Table 1 diagnostics-13-02249-t001:** Literature review of CHMCF with lung metastasis (21 cases).

Author (Year Published)	Maternal Age (years)	Pregnancy Type	GA at Diagnosis	BhCG Level at Diagnosis	Radiologic Finding of Complete Hydatidiform Mole	Detection of Pulmonary Metastasis	Pregnancy Outcome
Block and Merrill (1982) [16]	36	NS	On delivery	NS	Not obtained	Post OP	Amniotomy and delivery at 35 weeks
Jinno (1994) [17]	35	IVF	12 weeks	256,000 mIU/mL	Multiple cystic echoes	GA 17 weeks	Emergent cesarean section at 31 weeks (Infant died 4 h postpartum)
Osada (1995) [18]	30	Natural conception	24 weeks	478,000 mIU/mL	Typical molar pregnancy (four fifths)	7 weeks after delivery	Intrauterine fetal death and evacuation at 25 weeks
Ishii (1998) [19]	37	Natural conception	22 weeks	NS	NS	NS	Vaginal delivery at 40 weeks
Bruchium (2000) [20]	25	hMG/hCG	NS	35 MOM	Uterine wall mass	Post OP	Cesarean section at 26 weeks
Kashimura (2001) [21]	30	NS	13 weeks	684 ng/mL	Empty gestational sac with microcystic pattern	5 weeks after termination	Dilatation and evacuation (Termination)
Steigrad (2004) [22]	NS	NS	First trimester	NS	NS	Post OP	Cesarean section
Makary (2010) [23]	19	NS	25 weeks	228,000 mIU/mL	Large cystic mass	2 months after delivery	Emergent cesarean section at 25 weeks
Lee (2010) [24]	39	IVF-ET	13 weeks	1,307,693 mIU/mL	Diffuse vesicular pattern	Post OP	Hysterostomy (Termination)
Sasaki (2012) [8]	36	NS	15 weeks	440,000 mIU/mL	Typical classic molar pattern	GA 32 weeks	Spontaneous labor at 33 weeks
Sanchez-Ferrer (2013) [25]	28	Natural conception	11 weeks	395,000 mIU/mL	Multiple small cysts and a characteristic snowstorm pattern	Post OP	Suction curettage (Termination) at 13 weeks
Sanchez-Ferrer (2014) [26]	35	Natural conception	First trimester	963,971 mIU/mL	Mass of vesicular structures with snowstorm pattern	Post OP	Subtotal hysterectomy at 15 weeks (Termination and uterine rupture)
Peng (2014) [27]	34	NS	20 weeks	31,0277.7 mIU/mL	Multiple cystic spaces	4 months after delivery	Cesarean section at 37 weeks
Himoto (2014) [28]	34	Natural conception	9 weeks	1,124,200 mIU/mL	Multicystic lesion	Post OP	Artificial abortion (Termination)
Maeda (2018) [29]	33	NS	24 weeks	156,800 mIU/mL	Multicystic lesions	GA 29 weeks	Cesarean section and hysterectomy at 31 weeks
Nobuhara (2018) [30]	42	IVF	45 days	450,000 miU/mL	Subchorionic hematoma with multivesicular features	5 weeks after termination	Aspiration curettage at 9 weeks (Termination) and delayed hysterectomy
Odera (2019) [11]	34	NS	14 weeks	900,000 mIU/mL	Mixed cystic and solid lesion with internal vascularity	GA 23 weeks	Cesarean section at 23 weeks (Infant died a few hours postpartum)
Sindiani (2020) [31]	33	NS	13 weeks	171,820 mIU/mL	A sac filled with a complete molar pregnancy	Post OP	Hysterostomy (Termination)
Mok (2021) [32]	34	NS	10 weeks	free: 13.225 MoM	Multiple cystic area	Post OP	Emergent cesarean section at 32 weeks
Alpay (2021) [33]	33	ICSI	12 weeks	425,000 mIU/mL	Echogenic mass resembling molar placenta	8 weeks after delivery	Cesarean section at 26 weeks
Jung (2023) [This work]	41	Natural conception	Not done	NS	Not identified	Post OP	Cesarean section at 28 weeks

GA: gestational age, BhCG: beta-human chorionic gonadotropin, NS: not specified, hMG/hCG: human menopausal gonadotropin/human chorionic gonadotropin, IVF: in vitro fertilization, MoM: multiples of median, ET: embryo transfer, ICSI: intracytoplasmic sperm injection.

## Data Availability

All data are contained within the article.

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
