# Peer review of "Complete Hydatidiform Mole with Lung Metastasis and Coexisting Live Fetus: Unexpected Twin Pregnancy Mimicking Placenta Accreta"

_diagnostics, 2023, doi:10.3390/diagnostics13132249_

Round 1
Reviewer 1 Report
There were some similar or related reports and I am concerned about whether this article will be helpful to the readers. The article has no new ideas or original insights for the reader.For example, some similar reports can be listed:
1.Lu B, Ma Y, Shao Y, Xu E. Twin pregnancy with complete hydatidiform mole and co-existing fetus: A report of 15 cases with a clinicopathological analysis and DNA genotyping. Pathol Res Pract. 2022 Sep 6;238:154116.
2.Alpay V, Kaymak D, Erenel H, Cepni I, Madazli R. Complete Hydatidiform Mole and Co-Existing Live Fetus after Intracytoplasmic Sperm Injection: A Case Report and Literature Review. Fetal Pediatr Pathol. 2021 Oct;40(5):493-500.
N/A
Author Response
Comment 1. There were some similar or related reports and I am concerned about whether this article will be helpful to the readers. The article has no new ideas or original insights for the reader. For example, some similar reports can be listed:
1.Lu B, Ma Y, Shao Y, Xu E. Twin pregnancy with complete hydatidiform mole and co-existing fetus: A report of 15 cases with a clinicopathological analysis and DNA genotyping. Pathol Res Pract. 2022 Sep 6;238:154116.
2.Alpay V, Kaymak D, Erenel H, Cepni I, Madazli R. Complete Hydatidiform Mole and Co-Existing Live Fetus after Intracytoplasmic Sperm Injection: A Case Report and Literature Review. Fetal Pediatr Pathol. 2021 Oct;40(5):493-500.
Answer 1. I sincerely appreciate the reviewer’s comment. As pointed out, there has been various literature reports of complete hydatidiform mole with coexisting fetus (CHMCF). Compared to these previous studies, this case report demonstrates the importance of pathologic examination, and it is considered special that in the clinical suspicion of placenta accreta, coexisting complete hydatidiform mole with pulmonary metastasis could be diagnosed after cesarean section. Additionally, the literature review of this study was about CHMCF with lung metastasis and in the most cases, coexisting complete hydatidiform mole were diagnosed at first or second trimester, unlike this case report. Based on the clinical context, multiple lung metastases that could be harmful to the patient were diagnosed through the accurate pathological examination, so, in my opinion, this paper is worth reporting and can be helpful for obstetricians and pathologists to diagnose and manage this disease properly. I made corrections and clarifications as the reviewer suggested (page 5, line 192-198).
Reviewer 2 Report
Well written case report. The authors report an unexpected twin pregnancy with complete hydatidiform mole and a coexisting fetus and extrauterine metastasis which were clinically mimicking placenta accreta. According to this case, we must consider that, such uncommon cases can be detected by pathological examinations, so it should always be conducted with caution even in usual specimen. And if placenta accreta is suspected, it is worth considering a serum HCG test check when only limited clinical information is available, such as in this case. Graphics, pathologic figures and radiologic imagines are very good qualified. I think this paper can be acceptable.
Good english.
Author Response
Comment 1. Well written case report. The authors report an unexpected twin pregnancy with complete hydatidiform mole and a coexisting fetus and extrauterine metastasis which were clinically mimicking placenta accreta. According to this case, we must consider that, such uncommon cases can be detected by pathological examinations, so it should always be conducted with caution even in usual specimen. And if placenta accreta is suspected, it is worth considering a serum HCG test check when only limited clinical information is available, such as in this case. Graphics, pathologic figures and radiologic imagines are very good qualified. I think this paper can be acceptable.
Answer 1. I sincerely appreciate your thoughtful comments and review. Through the accurate comments made by the reviewer, I better understand the critical issues in the paper and improve the report.
Reviewer 3 Report
This case report describes the outcome of a complete hydatidiform mole with a coexisting alive fetus. The case report itself is briefly presented and the author discusses the outcome of these case with a narrative review.
The manuscript is well structured but is difficult to read at times and must undergo formal language editing.
Specific comments:
1. What was the obstetric/antenatal history in this case. As the reader is left with the question why this diagnosis was not made earlier during the routine second trimester ultrasound evaluation.
2. Why was placenta accreta suspected with only the appearance of hypervascularity. Were there any other ultrasound findings suggestive of placenta accreta? Did she have any previous caesarean sections or uterine interventions that could influence her risk profile to jump to this conclusion?
3. Intraoperatively what exactly was found. It is stated in the manuscript that the placenta was manually removed but was there any FIGO criteria met to make the diagnosis of intraoperative placenta accrete?
4. Table 1: With NA do you mean not specified (NS).
The manuscript is well structured but is difficult to read at times and must undergo formal language editing.
Author Response
Comment 1. What was the obstetric/antenatal history in this case. As the reader is left with the question why this diagnosis was not made earlier during the routine second trimester ultrasound evaluation.
Answer 1. I sincerely appreciate the reviewer’s valuable comment. The patient had a history of dilatation and evacuation due to spontaneous abortion at GA (gestational age) 11 weeks four years ago. And she delivered a 3.2 kg weight-female baby at GA 41 weeks 3 years ago. In this pregnancy, she was confirmed to be pregnant while living abroad but has a history of entering South Korea during second trimester due to the pandemic COVID-19 virus (Patient’s delivery date: May 09, 2022). The limitations of hospital visits during pandemic period of COVID-19 may be considered as a reason for the delay in the prenatal diagnosis of CHMCF. And significant amount of necrosis can also be another factor that made diagnosis difficult. I added explanations for the corresponding history as the reviewer suggested (page 2, line 43-45; page 5, line 162-168).
Comment 2. Why was placenta accreta suspected with only the appearance of hypervascularity. Were there any other ultrasound findings suggestive of placenta accreta? Did she have any previous caesarean sections or uterine interventions that could influence her risk profile to jump to this conclusion?
Answer 2. I sincerely appreciate the reviewer’s valuable comment. Other findings also included bridging vessels and multiple irregular lacunae within the placenta in the color Doppler ultrasound. Previous history of evacuation (mentioned in Answer 1), maternal age (41 years old), and ultrasonographic findings suggested placenta accreta. I made corrections and additional information as the reviewer suggested (page 2, line 51-55).
Comment 3. Intraoperatively what exactly was found. It is stated in the manuscript that the placenta was manually removed but was there any FIGO criteria met to make the diagnosis of intraoperative placenta accrete?
Answer 3. I sincerely appreciate the reviewer’s comment. As ultrasound findings suggested placenta accreta, the obstetrician obtained informed consent for cesarean section with the possibility of uterine artery embolization and hysterectomy in case of excessive bleeding. Intraoperatively, the uterus shows no obvious distension over the placental bed and surface is clear without gross neovascularity. After initial trial of manual removal for mildly adherent placenta, bleeding was present but controlled after intravenous Pitocin (10 unit) injection. So, no further procedure was proceeded. As the reviewer pointed out, it is not fully sufficient diagnostic factors of FIGO criteria for placenta accreta spectrum (PAS). However, preoperative ultrasound (mentioned in Answer2) and experienced clinician’s suspicion did not exclude the placenta accreta. I made corrections and clarifications as the reviewer suggested (page 2, line 66-75).
Comment 4. Table 1: With NA do you mean not specified (NS).
Answer 4. I sincerely appreciate the reviewer’s comment. Yes, I made corrections and clarifications as the reviewer suggested (page 6-7, table 1).
Comment 5. The manuscript is well structured but is difficult to read at times and must undergo formal language editing.
Answer 5. I sincerely appreciate the reviewer’s suggestion and overall corrections have been made.
Round 2
Reviewer 1 Report
1. Some of the literature review of CHMCF with lung metastasis had many not specified content. (For example Matsui (2000) all of the 4 cases and Lu (2022) all of the 5 cases had some not specified content including Pregnancy type, GA at diagnosis, HCG level at diagnosis, Radiologic finding of complete hydatidiform mole and Detection of pulmonary metastasis. We suggested to add relevant contents or delete relevant cases.
2. We had readed the article Matsui, H. et al. Hydatidiform mole coexistent with a twin live fetus: a national collaborative study in Japan. Hum Reprod. 2000, 15, 608-611 carefully. We found that there were 18 patents with complete hydatidiform mole coexistent with fetuses not 4 cases, please check the relative content.
Author Response
Comment 1. Some of the literature review of CHMCF with lung metastasis had many not specified content. (For example Matsui (2000) all of the 4 cases and Lu (2022) all of the 5 cases had some not specified content including Pregnancy type, GA at diagnosis, HCG level at diagnosis, Radiologic finding of complete hydatidiform mole and Detection of pulmonary metastasis. We suggested to add relevant contents or delete relevant cases.
Answer 1. I sincerely appreciate the reviewer’s comment. I deleted 9 cases in the table as the reviewer suggested (page 6-7, table 1).
Comment 2. We had readed the article Matsui, H. et al. Hydatidiform mole coexistent with a twin live fetus: a national collaborative study in Japan. Hum Reprod. 2000, 15, 608-611 carefully. We found that there were 18 patents with complete hydatidiform mole coexistent with fetuses not 4 cases, please check the relative content.
Answer 2. I sincerely appreciate the reviewer’s comment. As mentioned in discussion (page 5, line 178-183), the literature review was collected by searching for ‘complete hydatidiform mole and coexisting fetus (CHMCF) with lung metastasis’. In 18 cases Matsui et al. reviewed in 2000, only 6 cases (case no. 5, 6, 11, 14, 15 and 17) had pulmonary metastasis. And two of them (case no. 15 and 17) were originally presented in studies by Osada et al. [A complete hydatidiform mole coexisting with a normal fetus was confirmed by variable number tandem repeat (VNTR) polymorphism analysis using polymerase chain reaction. Gynecol Oncol 1995;56:90-3] and Ishii et al. [Genetic differentiation of complete hydatidiform moles coexisting with normal fetuses by short tandem repeat–derived deoxyribonucleic acid polymorphism analysis. American Journal of Obstetrics and Gynecology. 1998, 179, 628-634], each (included in table 1). Other 4 cases of Matsui et al. are now deleted in revised version by the reviewer ’s suggestion (mentioned in Answer 1) (page 6-7, table 1).